# Chain of Thought Imitation with Procedure Cloning

**Mengjiao Yang**
Google Brain, UC Berkeley
sherryy@google.com

**Dale Schuurmans**
Google Brain, Unversity of Alberta
schuurmans@google.com

**Pieter Abbeel**
UC Berkeley
pabbeel@cs.berkeley.edu

**Ofir Nachum**
Google Brain
ofirnachum@google.com

## Abstract

Imitation learning aims to extract high-performance policies from logged demonstrations of expert behavior. It is common to frame imitation learning as a supervised learning problem in which one fits a function approximator to the input-output mapping exhibited by the logged demonstrations (input *observations* to output *actions*). While the framing of imitation learning as a supervised input-output learning problem allows for applicability in a wide variety of settings, it is also an overly simplistic view of the problem in situations where the expert demonstrations provide much richer insight into expert behavior. For example, applications such as path navigation, robot manipulation, and strategy games acquire expert demonstrations via planning, search, or some other multi-step algorithm, revealing not just the output action to be imitated but also the procedure for how to determine this action. While these intermediate computations may use tools not available to the agent during inference (e.g., environment simulators), they are nevertheless informative as a way to explain an expert's mapping of state to actions. To properly leverage expert procedure information without relying on the privileged tools the expert may have used to perform the procedure, we propose *procedure cloning*, which applies supervised sequence prediction to imitate the series of expert computations. This way, procedure cloning learns not only *what* to do (i.e., the output action), but *how* and *why* to do it (i.e., the procedure). Through empirical analysis on navigation, simulated robotic manipulation, and game-playing environments, we show that imitating the intermediate computations of an expert's behavior enables procedure cloning to learn policies exhibiting significant generalization to unseen environment configurations, including those configurations for which running the expert's procedure directly is infeasible.[1]

'

## 1 Introduction

The idea of learning by imitation in autonomous agents closely resembles how humans (especially children) learn in real life — by watching and mimicking how someone else performs a certain task [1]. While humans are able to generalize exceedingly well from a small number of demonstrations, today's imitation-learned autonomous agents often struggle in situations that only slightly differ from the demonstrations, e.g., opening a door in a different shape or color [2]. One explanation for such a difference is that while humans imitate, we *understand* the task at a high level as opposed to only remembering a mapping from images to actions [3], and indeed, generalization failures can also occur

---

[1] https://github.com/google-research/google-research/tree/master/procedure_cloning.

36th Conference on Neural Information Processing Systems (NeurIPS 2022).

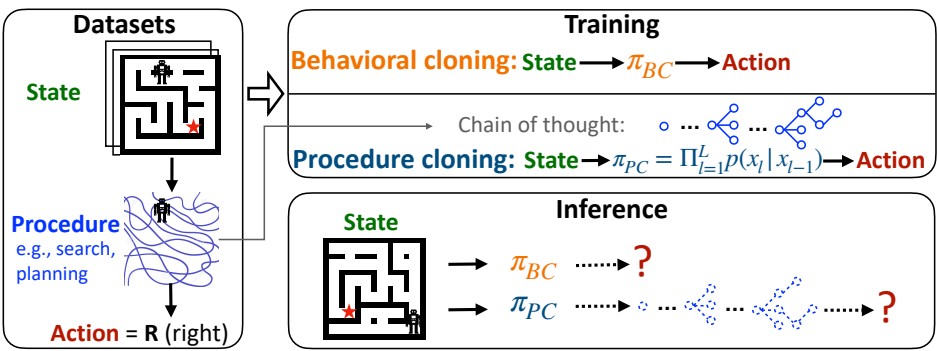

Figure 1: Visualization of the dataset collection, training, and inference of BC and PC on a maze navigation task. During dataset collection, the expert uses a search procedure to determine the optimal action to generate a path to the goal location (red star). During training, BC discards these intermediate search outputs and learns to map states to actions directly. In contrast, PC learns the complete sequence of intermediate computations (i.e., branches and backtracks) associated with the search procedure. During inference, PC generates a sequence of intermediate search outcomes emulating the search procedure on a new test map before outputting the final action.

in humans when there is a lack of understanding of the underlying reasons for the behavior, such as solving a complex math problem. As a result, students are told to "learn principles, not formulas" and "understand, do not memorize" to encourage better generalization — to be able to solve problems that are similar but different from the ones taught in lectures [4, 5, 6]. Just as students are taught the step-by-step derivations of a math problem during a lecture, is it possible to have an equivalent "chain of thought" supervision for training autonomous agents?

While solving complex math derivations may not be the predominant application of today's imitation-learned autonomous agents, the tasks they are commonly applied to — e.g., path navigation, robot manipulation, and strategy games — often *do* employ chain-of-thought reasoning procedures in the form of planning, search, or other multi-step algorithms when collecting expert data [7, 8, 9]. Since such multi-step algorithms are application-specific and therefore lack a common representation which can be systematically characterized, it is typical for imitation learning to assume access to only logged demonstrations of state-action pairs, leaving out the much richer insight into expert behavior provided by the algorithm's intermediate computations. In addition, the planning or search procedures used by the expert may rely on tools not available to the agent during inference (e.g., environment simulators), so how these intermediate computations should be used to facilitate imitation learning is not clear.

In this work, we formulate *chain of thought imitation* as an extension of traditional imitation learning. Different from the traditional imitation learning setup where an agent only has access to expert state-action pairs as demonstrations, chain of thought imitation also has access to the intermediate computations that generated the expert state-action pairs in the training data (Figure 1). We then propose *procedure cloning* (PC), an alternative to behavioral cloning (BC) [10], which applies supervised sequence prediction to imitate the complete series of of expert computations before outputting an expert action (Figure 1). Procedure cloning learns a policy by maximizing the likelihood of the joint distribution of procedure observations and expert actions, which can be modeled autoregressively using a transformer-like architecture [11]. During inference, a procedure cloned policy autoregressively generates procedure observations from a given input state, mimicking the computations of a search or planning algorithm before outputting the final action, thus avoiding any reliance on privileged tools or information used by the expert's procedure. From a modeling perspective, procedure cloning can employ a more expressive model (e.g., transformer) trained on more data (i.e., procedure observations), which leads to better generalization according to the new "scaling law" of large language models [12]. Intuitively, procedure cloning learns not only what to do (i.e., the output action), but how and why to do it (i.e., the procedure), which further resembles how humans learn complex tasks.

We demonstrate how to conduct procedure cloning and leverage the generalization ability of procedure cloned agents on a variety of path navigation, robotic manipulation, and strategy game tasks. For example, in path navigation an expert trajectory may be determined by using a BFS search algorithm on an annotated map (e.g., $x, y$ coordinates of obstacles), which are expensive to obtain [13]. We

show that a procedure cloning agent can successfully learn to imitate BFS on a previously unseen test map without requiring additional annotations, achieving 100% test accuracy while a BC agent completely fails to navigate (0% accuracy) when the maze layout changes. Similarly, in an image-based robotic manipulation task, we observe that BC quickly overfits to the set of training images, whereas procedure cloning learns to predict the intermediate computation outcomes of a scripting policy, thus generalizing much better and achieving a success metric of 83.9% compared to 78.2% from the previous state-of-the-art [14]. Finally, in strategy games expert trajectories are collected by running MCTS [15], which requires access to a simulator and can be extremely slow [16]. We show that procedure cloning can effectively learn from the path traversed by MCTS collected in a deterministic environment to then successfully generalize in a zero-shot manner to stochastic environments and more difficult game settings, where running MCTS, even if given access to a simulator, performs poorly.

## 2 Related Work

**Generalization in sequential decision making.** Learning decision making policies with good generalization properties has a long-standing history in bandit [17, 18, 19], imitation learning [20], and reinforcement learning (RL) [21, 22, 23, 24] settings in the form of regret or Probably Approximately Correct (PAC) bound analysis. These studies are concerned with generalization in a continual learning setup without explicit separation of training and testing stages. We are more interested in an agent's ability to generalize to a separate "test" environment whose configuration is different from the training environment, which is often what happens when an agent is deployed to the real world after being trained in a simulator. With high-capacity neural network parametrized policies being the norm in training image-based agents, the risk of overfitting to the training environment is nontrivial [25, 26]. Various works have injected stochasticity into the training process including using stochastic policies [27], random starts [28, 29], sticky actions [30], and frame skipping [31] as a means to prevent overfitting to the specific environment dynamics experienced by the agent during training. A variety of image-based regularization techniques have also been applied to deep RL including dropout and $l$-2 regularization [32], data augmentation and batch normalization [33], domain randomization [34], and network randomization [35]. Instead of improving generalization through regularization or data augmentation like in existing work, we instead ask the question of whether learning the sequence of computations as opposed to only the final expert action during training can help an agent generalize better.

**Access to additional task information** Many previous works in imitation learning and RL have observed that access to additional task information can help an agent learn better policies. For instance, [36] relies on access to the simulator for collecting more expert trajectories to reduce the quadratic error in task horizon to linear. [37] assumes that expert demonstrations contain both high-level and low-level trajectories, and learns a hierarchical policy explicitly. While our procedure observations can appear to be high-level labels or sub-goals similar to [37, 38], we do not make assumptions about the structure of procedure observations, which can simply be scalar variable values computed during program execution. There is also a large body of literature that assumes access to a suboptimal offline dataset in addition to the expert demonstration data collected from the same environment, and conduct representation learning [39, 40, 41, 42, 43], hierarchical skill extraction [44, 45, 46], or dynamics model learning [47, 48, 49] on the suboptimal offline data followed by imitation learning from an expert. These works commonly assume good coverage in the offline data, which requires large amounts of state-action pairs being collected from running additional policies in the environment. We instead propose observing additional information from only running the expert policy, which requires far fewer state-action pairs being collected for training. Another line of existing work in the direction of multi-task learning suggests that learning an auxiliary task enhances imitation learning or RL [50, 51, 2, 52, 2, 53, 54]. For instance, [55] showed that predicting internal features of the emulator (e.g., enemy on the screen) is beneficial, [56] found predicting scene depth helps navigation, and [57] found predicting explanations helps relational tasks with causal structure. Chain of thought imitation differs from multi-task learning in that procedures directly influence the action outcomes through computations, in which case learning the procedure directly (as opposed to treating it as an auxiliary objective similar to existing work) is more beneficial. The graphical model in Figure 2 visualizes this distinction.

**Chain of thought sequence modeling.** The idea of decomposing multi-step problems into intermediate steps (the so-called chain of thought [58]) and learning the intermediate steps using a sequence model has been applied to *domain specific* problems such as program induction [59], learning to

solve math problems [60], learning to execute [61], learning to reason [62, 63, 64, 65, 66, 67], and language model prompting [58]. The chain of thought imitation learning problem we formulate is *domain agnostic* and applicable to many sequential decision making task traditionally solved by imitation learning in a Markovian setting such as robot locomotion, navigation, manipulation, and strategy games. Unlike language-based tasks, problems in decision making have only recently started being explored by language models [68, 69, 70, 71, 72, 73], as the Markovian nature of these problems brings the value of sequence modeling into question. Our work contributes to bridging the gap between learning memoryless policies in Markovian environments and the intuition that large sequence models should help in reasoning-based decision making.

## 3 Preliminaries

In this section, we define relevant notations and introduce the problem of imitation learning. We also review behavioral cloning (BC), which our work aims to improve.

**MDP notations.** Consider a Markov decision process (MDP) [74] $\mathcal{M} := \langle S, A, \mathcal{R}, \mathcal{T}, \mu, \gamma \rangle$, consisting of a state space $S$, an action space $A$, a reward function $\mathcal{R} : S \times A \to \mathbb{R}$, a transition function $\mathcal{T} : S \times A \to \Delta(S)^2$, an initial state distribution $\mu \in \Delta(S)$, and a discount factor $\gamma \in [0, 1)$. A policy $\pi : S \to \Delta(A)$ interacts with the environment starting at an initial state $s_0 \sim \mu$. At each timestep $t \geq 0$, an action $a_t \sim \pi(s_t)$ is sampled and applied to the environment, and the environment transitions into the next state $s_{t+1} \sim \mathcal{T}(s_t, a_t)$ while producing a scalar reward $\mathcal{R}(s_t, a_t)$. The state visitation distribution $d^\pi(s)$ induced by a policy $\pi$ is defined as $d^\pi(s) := (1 - \gamma) \sum_{t=0}^{\infty} \gamma^t \cdot \Pr[s_t = s | \pi, \mathcal{M}]$. We use $(s, a) \sim d^\pi$ to denote $s \sim d^\pi, a \sim \pi(s)$.

**Imitation learning.** In the standard imitation learning setting, $\mathcal{R}, \mathcal{T}$, and $\mu$ are unknown to the agent. Learning solely takes place on a fixed dataset of expert state-action pairs $\mathcal{D}_* = \{(s^{(i)}, a^{(i)})\}_{i=1}^n$ collected by an expert policy $\pi_*$ by sampling $s^{(i)} \sim d^{\pi_*}$ and $a^{(i)} \sim \pi_*(s^{(i)})$. The goal of imitation learning is to train a policy $\pi : S \to \Delta(A)$ on $\mathcal{D}_*$, such that $\pi$ closely approximates $\pi_*$ according to some metric such as the Kullback–Leibler (KL) divergence of their state visitation distributions $D_{\mathrm{KL}}(d^{\pi_*}(s) \| d^\pi(s))$. Regardless of the choice of this metric, it is computed over states including those outside of the training data $\mathcal{D}_*$, and requires an agent to generalize over unseen states to achieve good test performance.

**Behavioral cloning (BC).** The nature of learning from an expert dataset $\mathcal{D}_*$ leads to the common framing of imitation learning as a supervised problem of learning state to action mappings. Under this framing, behavioral cloning (BC) [10] proposes to solve the imitation learning problem by learning $\pi$ that minimizes

$$J_{\mathrm{BC}}(\pi) := \mathbb{E}_{(s,a) \sim (d^{\pi_*}, \pi_*)}[-\log \pi(a|s)] \tag{1}$$

using samples from the training data $(s, a) \sim \mathcal{D}_*$. Equation 1 can also be viewed as the cross entropy loss for multi-class classification of state to action mappings. Agents trained with this vanilla BC objective (i.e., direct max-likelihood on the training dataset $\mathcal{D}_*$) can quickly fail to generalize to unseen states when the expert data size is small compared to the state-action space of the environment [36, 75], which is often the case when the states are based on images. Image-based inputs also pose the challenge of learning the right invariances for the agent to generalize from training to testing environments with different configurations (e.g., maze layouts and environment stochasticity).

## 4 Procedure cloning

In this section, we first observe that in many imitation learning situations, expert demonstrations can provide much richer insight into the desirable behavior than just the final optimal action. We formulate learning under these situations as *chain of thought imitation*, where an agent can learn from not just the optimal action but the "thought process" an expert goes through before arriving at the final decision. We then formalize such thought process as *procedures*, and propose *procedure cloning* for learning such thought process through supervised sequence prediction.

### 4.1 Chain of thought imitation

Depending on the form of the expert policy $\pi_*$ used to generate the training data $\mathcal{D}_*$, an agent potentially has access to a rich set of information about a task which can facilitate learning. For

---

[2]$\Delta(\mathcal{X})$ denotes the simplex over a set $\mathcal{X}$.

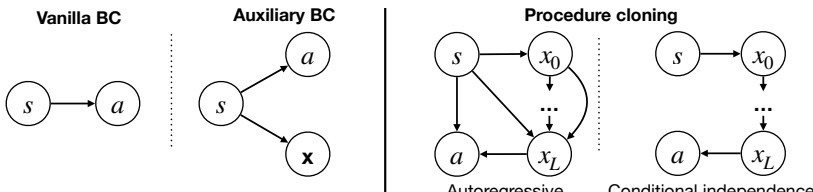

Figure 2: Graphical models of vanilla BC, auxiliary BC, and procedure cloning with autoregressive and conditionally independent factorization. Node $s$ represents an input MDP state, $a$ represents an expert action, and $\mathbf{x}$ represents the sequence of procedure observations $(x_0, ..., x_L)$.

instance, when $\pi_*$ is some scripting policy following a fixed set of rules [76], the rules (e.g., "if close to enemy then fire") reveal causal information between firing and the enemy disappearing. In other settings when $\pi_*$ is a search algorithm, the induced search tree exposes the path for finding the optimal action. When $\pi_*$ is a multi-step hand-coded algorithm, the breakdown of the steps (e.g., "first move to objects then sweep") reveals important ordering information about the task. Even in the case where $\pi_*$ is a human demonstrator, we can ask the human to explain the thought process that led to their decision. We refer to learning from the procedures (e.g., planning, search, multi-step algorithm) that generated the final action as *chain of thought imitation*.

## 4.2 Procedures and procedure observations

To formulate chain of thought imitation as a learning problem, we define a procedure $\mathbf{\Pi} : S \to \Delta(A)$ as a sequence of computations $(\Pi_0, \Pi_1, ..., \Pi_L, \Pi_{L+1})$ that first transform an input state $s \in S$ into some computation state (e.g., a variable value) using $\Pi_0 : S \to \mathbb{R}^d$, followed by repeatedly applying each subprocedure, $\Pi_\ell : \mathbb{R}^d \to \mathbb{R}^d, \forall \ell \in [1, L]$, to the computation state before mapping the last computation state back to the MDP action space using $\Pi_{L+1} : \mathbb{R}^d \to \Delta(A)$, in a sequential order. A procedure can be broken down to subprocedures at any user-defined granularity (e.g., a function or a for loop), depending on how frequently the computation state is retrieved for procedure learning. Each pair of training data, $(s, a) \sim \mathcal{D}_*$, is acquired through executing such a procedure, i.e., $\pi_* = \mathbf{\Pi}$.

We define *procedure observations* as the sequence of computation states captured from a procedure: $\mathbf{x} = (x_0, ...x_L) \in \mathbb{R}^{(L+1) \times d}$ where $x_0 = \Pi_0(s)$ and $x_\ell = \Pi_\ell \circ \Pi_{l-1} \circ ... \circ \Pi_1 \circ \Pi_0(s), \forall \ell \in [1, L]$. The training data with procedure observations is denoted as $\mathcal{D}_\Pi = \{(s^{(i)}, \mathbf{x}^{(i)}, a^{(i)})\}_{i=1}^n$. Note that we do not assume any procedure or procedure observation is available at test time, hence we still need to learn a policy $\pi$ that only takes $s$ as input.

## 4.3 Procedure cloning

With the procedure observations defined above, we are now ready for learning procedures through *procedure cloning* (PC). We model a PC policy by estimating the joint distribution of the procedure observations and the final action conditioned on the input state $p(a, \mathbf{x}|s)$, which we can factorize autoregressively as:

$$p(a, \mathbf{x}|s) = p(a|\mathbf{x}, s) \cdot \Pi_{l=1}^L p(x_\ell|\mathbf{x}_{<\ell}, s) \cdot p(x_0|s). \tag{2}$$

Under this factorization, estimating $p(a, \mathbf{x}|s)$ reduces to estimating each conditional factor, which can be parametrized using a transformer model [11]. The autoregressive factorization is highly flexible, but if the amount of expert demonstrations is small, and each procedure observation $x_\ell$ only depends on the previous procedure observation $x_{\ell-1}$ (i.e., the computation states are fully observed), and the final action only depends on the last computation state, a conditionally independent factorization can be more desirable. In other words, autoregressive models need more data to train, so the conditional independence factorization below is preferable if the procedure information available fully captures the computation state:

$$p(a, \mathbf{x}|s) = p(a|x_L) \cdot \Pi_{l=1}^L p(x_\ell|x_{\ell-1}) \cdot p(x_0|s). \tag{3}$$

The graphical models of the two factorizations of PC policies are shown in Figure 2. To learn a PC policy, we maximize the empirical likelihood of the joint distribution in Equation 2 on given samples

$\mathcal{D}_\Pi = \{(s^{(i)}, \mathbf{x}^{(i)}, a^{(i)})\}_{i=1}^n$:

$$\min_{\phi, \theta, \psi} J_{\text{PC}}(\phi, \theta, \psi) = \hat{\mathbb{E}}_{(s, \mathbf{x}, a) \sim \mathcal{D}_\Pi}[-\log p(a, \mathbf{x}|s)] \tag{4}$$

$$= \hat{\mathbb{E}}_{(s, \mathbf{x}, a) \sim \mathcal{D}_\Pi}\left[-\log q_\psi(a|\mathbf{x}, s) - \sum_{\ell=1}^{L} \log p_\theta(x_\ell|\mathbf{x}_{<\ell}, s) - \log p_\phi(x_0|s)\right]. \tag{5}$$

**Connection to BC with auxiliary tasks.** The PC objective in Equation 5 can be reduced to the vanilla BC objective in Equation 1 by discarding the second and third term inside the expectation of Equation 5 and setting $q_\psi(a|\mathbf{x}, s) = \pi(a|s)$. We note that several previous works [51, 50, 2, 52, 55, 56] have used procedure information as auxiliary tasks to BC, which may be interpreted as learning $p(a, \mathbf{x}|s)$ under the assumption that $a$ and $\mathbf{x}$ are independent conditioned on $s$: $p(a, \mathbf{x}|s) = \pi(a|s) \cdot p(\mathbf{x}|s)$. Procedure cloning instead focuses on situations where such a conditional independence assumption does not hold (i.e., $a$ is directly computed by the procedure represented by $\mathbf{x}$), and in these situations, as we will show in our experiments, treating the procedure information as a precursor to $a$ can perform better than using it as an auxiliary task. The graphical models of vanilla BC and BC with auxiliary task objective are shown in Figure 2.

## 5 Proof of concept: Synthetic maze navigation

In this section, we study a tabular maze navigation task with synthetically generated maze layouts (see Figure 3). We describe the task setup, followed by how to extract procedure data from a breadth-first search (BFS) path planning algorithm to train a procedure cloning agent, and empirically show that procedure cloning indeed generalizes much better to unseen maze layouts than other BC baselines. While this proof of concept illustration might seem domain-specific to BFS in mazes, we will see in Section 6.3 that procedure cloning can be applied to many types of search or multi-step algorithms.

**Task description and evaluation protocol.** We use a gridworld maze environment in which an agent seeks to navigate to a goal location in a maze from a random starting location using 4 discrete actions including up (U), down (D), left (L), and right (R). The input to the agent is a multi-channel "image" with the maze wall, goal location, and agent location encoded in separate channels. The maze layout is algorithmically generated with random internal walls that form a tunnel-shaped map (see example maze in Figure 4 and details in Appendix A.1). We generate a set of mazes $S_0 \subset S$ and split $S_0$ into disjoint training $S_0^{\text{train}}$ and testing $S_0^{\text{test}}$ sets. We then generate expert trajectories by running BFS on only the training set of mazes $S_0^{\text{train}}$. At test time, the agent is evaluated on $S_0^{\text{test}}$ by their success rate of navigating to the goal. Figure 3 visualizes the data and training pipeline.

**Procedure data collection.** BFS is a common path planning algorithm for navigation [77, 78, 79, 80], which we use to generate expert trajectories for training an imitation learning agent. To compute the optimal action at each time step, BFS keeps track of a visited 2D array (colored cells in Figure 3) that marks whether each position (1) has been visited by the search, (2) if so, which action visited it, and (3) has a position been backtracked. We simply take a snapshot of the entire visited array as procedure observations $x_\ell$ every time BFS expands the search perimeter, resulting in a series of procedure data $\mathbf{x} = (x_1, ..., x_L)$ as shown in Figure 3. $\Pi_0$ is the identity map and $x_0 = s$. See pseudocode for collecting procedure observations in Appendix A.2.

**Procedure learning.** Since each of the visited 2D arrays $x_\ell$ only depends on the previous visited array $x_{\ell-1}$, and the final visited array after backtracking uniquely identifies the expert action on its own, we choose the conditionally independent factorization of $p(a, \mathbf{x}|s)$ (Equation 3) described in Section 4.3. Specifically, we parametrize $p_\theta(x_\ell|x_{\ell-1}), \forall \ell \in [1, L]$ using a deep convolutional neural network that takes in the current visited array $x_{\ell-1}$ as input and produces the next visited array $x_\ell$ as output. We optimize $\theta$ using the cross-entropy loss between the predicted and true next visited array. Since the procedure observations $x_\ell$ and the original input $s$ are in the same image space, $\phi(x)$ shares the same parameters as $p_\theta(x_\ell|x_{\ell-1})$. During inference when a new test maze layout $s$ is given, we apply $\hat{x}_0 = \phi(s)$ and $\hat{x}_\ell \sim p_\theta(\cdot|\hat{x}_{\ell-1})$ repeatedly until an array $\hat{x}_L$ is predicted for which the entry in $\hat{x}_L$ corresponding to the agent's current location is labelled as "backtracked", and we return the backtracked action as the final output action.

**Generalization results.** We compare procedure cloning to applying data augmentation with random crop, translation, and zoom (Aug BC) or auxiliary objective of predicting the visited array (Aux BC) to vanilla BC. BC policies are parametrized with convolutional neural networks (CNN) and multi-layer perceptrons (MLPs) (see hyperparameters in Appendix A.5). Aux BC receives the same

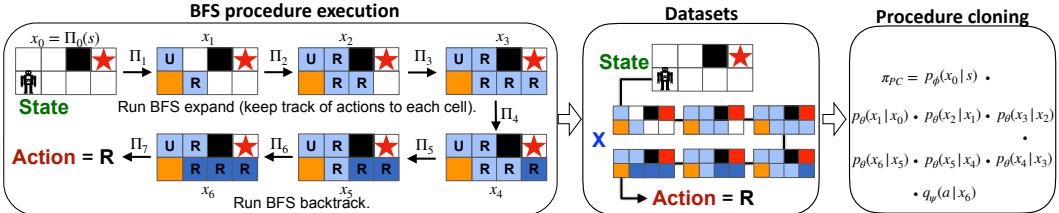

Figure 3: In a discrete maze, the expert employs BFS by first expanding a search perimeter until it encounters the goal cell, at which point it backtracks to find the optimal action at the starting state (cells in light blue are visited and dark blue are backtracked). We encode this algorithm as a sequence of procedure observations $(x_0, ..., x_6)$ of the intermediate computation states, with each $x_i$ represented by a 2D array and each cell of the array containing BFS-relevant information (i.e., whether this cell is being expanded or backtracked and the action recorded when expanding to this cell). Procedure cloning is trained to predict the entire sequence of computations from input state to output action using a sequential model $p(a|x_L) \cdot \Pi_{l=1}^L p(x_\ell|x_{\ell-1}) \cdot p(x_0|s)$.

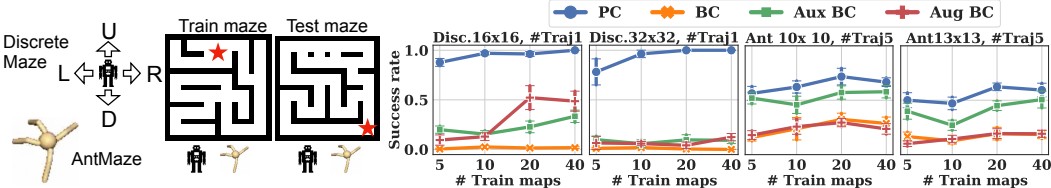

Figure 4: [Left] Visualization of the discrete maze (4 discrete actions) and AntMaze (8 continuous actions). [Right] Average success rate of PC and BC agents navigating to the goal from random start locations over 10 test mazes. Agents are trained on 5, 10, 20, 40 mazes of 1 and 5 expert trajectories on discrete maze and AntMaze, respectively. We find that procedure cloning leads to much better test maze generalization compared to alternative approaches.

information as PC when computing the auxiliary loss, i.e., the visitation maps are given to both PC and Aux BC. Figure 4 (Disc.$16 \times 16$ and Disc.$32 \times 32$) shows the average success rate (over 5 trajectories) of reaching the goal from random start locations on 10 test maze layouts unseen during training. Procedure cloning successfully generalizes to test mazes, whereas vanilla BC completely fails to learn (0% sucess rate) in bigger maze $32 \times 32$. Aux BC and Aug BC help in the smaller maze but not in the bigger maze. The poor test performance of BC is due to generalization failure as opposed to insufficient model capacity as BC's success rate on the training mazes are close to 100% (see Figure 10 in Appendix B.2). We also include additional experiments using VIN [81] as a baseline in Appendix B.1, showing that even an approach specialized to the structure of gridworld tasks exhibits poor generalization.

## 6 Experiments

We now evaluate procedure cloning in larger scale settings on tasks including simulated robotic navigation [82] and manipulation [83, 14], and learning to play MinAtar [84] (a miniature version of Atari [85]). Procedure cloning exhibits significant generalization to previously unseen maze layouts, positions of objects being manipulated, and environment configurations such as transition stochasticity and game difficulty in each of the tasks, respectively. See Appendix B for more results.

### 6.1 Evaluating continuous robot navigation in AntMaze

**Task description.** Following the discrete maze navigation task in Section 5, we now consider a more realistic setting of mimicking real-world robotic navigation. We adopt the AntMaze environment from D4RL [82], where an 8-DoF "Ant" quadruped robot with continuous state and action spaces is placed in a 2D maze environment. The original task designed for offline RL only has one maze layout which is not revealed to the agent (the agent only sees its current position and joint-based state measurements). To adopt the task for evaluating generalization across maze layouts, we also pass the algorithmically generated maze layouts from Section 5 to the agent as inputs. During evaluation, the agent needs to navigate to the goal in previously unseen maze layouts.

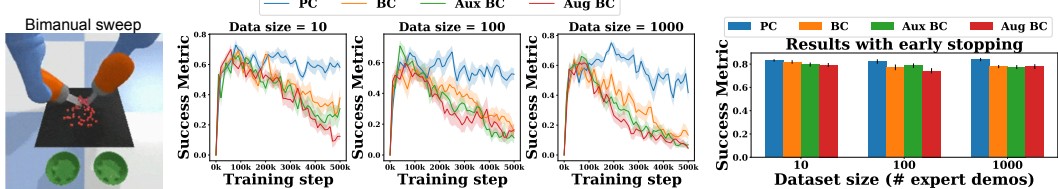

Figure 5: [Left] Visualization of the bimanual sweep task. [Middle] Average success metric (proportion of particles in bowls at the end of the episode) of PC and BC agents completing the bimanual sweeping task after learning on 10, 100, 1000 expert trajectories; each variant is an aggregate of 10 runs. All of our algorithm implementations use the implicit loss function described in [14] for this task. [Right] When using 1000 expert demonstrations with early stopping, PC achieves 83.9% compared to 78.2% success of the existing state-of-the-art achieved by implicit BC.

**PC implementation.** The procedure that generated the expert training trajectories for goal-reaching in AntMaze involves a low-level PID controller that is good for navigating to local locations close to the robot and a high-level waypoint generator which uses BFS search to find the next local waypoint [82]. We apply procedure cloning to the high-level waypoint generator following the same steps as in discrete maze described in Section 5. The predicted waypoint is then passed to a Gaussian parametrized policy (optimized with max-likelihood) together with the agent's joint measurements. For BC and variants, we use a convolutional neural network to embed the maze layout into a fixed dimensional vector and concatenate the robot joint measurements together with the agent and goal locations to a Gaussian policy.

**Generalization results.** Figure 4 (Ant 10 × 10 and Ant 13 × 13) shows the average (over 5 trajectories) success rate of the robot reaching the goal from random starting locations on 10 test mazes. Procedure cloning consistently provides significant benefits over vanilla BC, whereas data augmentation on the maze layout and predicting the `visited` array as an auxiliary loss are less helpful. We know the poor test performance of BC (and other baseline methods) is more due to generalization failure as opposed to insufficient model capacity because the success rate on the training mazes between procedure cloning and auxiliary BC are similar (Figure 11 in Appendix B.3).

## 6.2 Evaluating image-based robot manipulation

**Task description.** The bimanual sweeping task [83, 14] requires two 7-DoF robot arms equipped with spatula-like end-effectors to sweep a pile of particles evenly into two bowls while avoiding dropping particles between the tips of the spatulas. The scripted oracle for collecting expert trajectories uses access to privileged information including object poses and contact points, which are not accessible at test time. Rather, only high-resolution (96 × 96) images in conjunction with end-effector positions and orientations are given to the imitation-learned agent during inference. Actions are end-effector positions for each robot arm (6 dimensions for each arm for a total of 12 dimensions for the action); for how to incorporate kinematic actions into this environment, we refer the reader to [83]. Random seeds determining the initial position of the particles are partitioned so that test-time configurations are not seen in training.

**PC implementation and results.** The scripted policy first scoops the particles by moving the spatulas to their computed geometric center and then moves the spatulas to a bowl before releasing the particles. At each state of a trajectory, we collect the Cartesian coordinates of one of these two goal points – geometric center of the particles or position of the bowl – depending on the expert's behavior mode at that state, and use these as procedure observations. We supervise the PC agent to first predict these coordinates, then predict the actions conditioned on the predicted coordinates and end-effector positions and orientations. For both BC and PC we parameterize log-likelihood using energy-based models, also known as an *implicit* loss, which is state-of-the-art for this task [14]. We compare PC to BC trained directly on image inputs and auxiliary BC which learns to predict oracle coordinates as an auxiliary objective. We see that the BC variants quickly overfit to the training set of observations, whereas PC generalizes much better (Figure 5, left). Despite early stopping (taking the maximum evaluation success rate over all training steps), PC still outperforms all of the BC variants (Figure 5, right), improving significantly over the state-of-the-art.

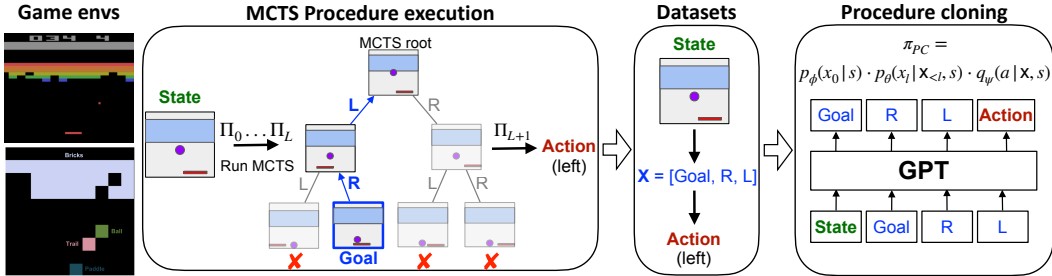

Figure 6: In the MinAtar game-playing environment, the expert uses MCTS $(\Pi_0, ..., \Pi_L)$ to find an optimal future trajectory [L, R, Goal]. We treat this future trajectory in reverse order [Goal, R, L] as procedure observations, so that procedure cloning is trained to first predict the goal image (MCTS leaf node) and then predict the optimal action sequence backwards from the goal using a GPT-like autoregressive model, ultimately predicting the expert's output action as its last prediction.

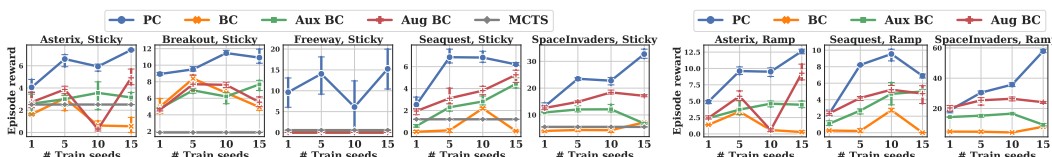

Figure 7: Average episode reward (over 50 episodes) of PC and BC agents playing MinAtar games over 3 test environments using sticky actions (left) and game difficulty ramping (right) not see in the training environments.

## 6.3 Evaluating strategy games in MinAtar

**Task description.** MinAtar is a minature version of the Atari Arcade Learning Environment [85] consisting of 5 games with simplified $10 \times 10$ multi-channel images as inputs. To generate the expert trajectories for training, we run a AlphaZero-style [15] Monte-Carlo tree search (MCTS) algorithm on the deterministic version of the environments to collect expert trajectories (see Appendix A.3 for details). During evaluation, we test the imitation-learned agents on a set of test environments with different seeds from the training environments where expert trajectories are collected. To further evaluate generalization, we apply sticky actions with probability 0.1 and game difficulty ramping to the test environments where such an option is available.

**PC implementation.** In contrast to running BFS in maze navigation where the `visited` array cleanly captures the entire state of the search, running MCTS in MinAtar is more convoluted, involving a number of MCTS simulation runs each with selection, expansion, rollouts, and backtrack steps and different tree structures, making capturing the full search state difficult. Fortunately, procedure cloning with autoregressive factorization (Equation 2) allows procedure data to be partial observations of the computation state, and so we elect to use only a subset of the MCTS computation states as supervision for PC. Namely, we record the optimal action sequence *after* the last MCTS simulation run (from the final search tree) and the goal image at the tree leaf as procedure observations (highlighted in blue in Figure 6); this is effectively the optimal future trajectory determined by MCTS. A PC policy is trained to use the input state to first predict the goal image using a CNN and then use the goal as input to an autoregressive action sequence model $p(x_\ell|\mathbf{x}_{<\ell}, s)$ where $x_\ell$ is the optimal action that is $\ell$-steps away from the goal (i.e., predicting the optimal action sequence *backwards* from the goal). Figure 6 illustrates the data and training pipeline of procedure cloning.

**Generalization results.** Figure 7 shows the average reward (over 50 episodes) collected by running PC and BC policies in 3 test environments with different environment seeds than the environments used to collect training trajectories. Figure 7 (left) evaluates generalization to stochastic environments, where we found sticky actions caused MCTS (gray) to struggle to search for good actions without extensive tuning. Figure 7 (right) evaluates generalization to more difficult game settings (available in 3 out of 5 MinAtar games). Autoregressive PC policy generalizes the best across all games and all settings.

# 7 Conclusion

We have identified a major gap between the imitation learning problem setup and how training data for imitation are often obtained in practice. Specifically, applications that use planning, search, or other multi-step algorithms reveal the procedure for determining an expert action in addition to outputting the expert action itself. This led us to formulate chain of thought imitation learning, where an agent is also given access to the intermediate computations that generated the expert state-action pairs. In order to generalize to test environments where intermediate computation is not available, we propose procedure cloning, which learns to predict the intermediate computation outcomes step-by-step using a sequence model, and emulates the intermediate computations during inference through autoregressive generation. Evaluation on a variety of navigation, manipulation, and game play tasks shows that procedure cloning generalizes much better to test environments of different configurations than alternative improvements over behavioral cloning. We hope the success of applying sequence modeling to imitation learning achieved by procedure cloning opens up new research directions in the intersection of large-scale sequence modeling and sequential decision making.

**Limitations.** One major limitation of procedure cloning compared to traditional BC is in the computational overhead, since PC needs to predict intermediate procedures. Furthermore, the choice of how to encode the expert's algorithm into a form amenable to PC is up to the practitioner. While we have presented ways to encode a variety of policies here (BFS, MCTS, scripted robotic policies), applying PC to other domains may require some amount of trial-and-error in designing the ideal computation sequence for PC.

## Acknowledgments and Disclosure of Funding

Thanks to Hanjun Dai, Kimin Lee, and Olivia Watkins for reviewing draft versions of this manuscript. Thanks to Hanjun Dai for discussions on the MinAtar task. Thanks to Pete Florence, Andy Zeng, and Jake Varley for assistance with the bimanual sweep task, and Justin Fu and Aviral Kumar for assistance with the AntMaze task.

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
