# OpenReview forum: "Chain of Thought Imitation with Procedure Cloning"
_NeurIPS.cc/2022/Conference — NeurIPS 2022 Accept_

### Official Review · Reviewer_4JKy · 2022-07-07

**Rating:** 7
**Confidence:** 4
**Soundness:** 3 good
**Presentation:** 4 excellent
**Contribution:** 3 good

**Summary:**

This paper seeks to improve traditional imitation learning, by exploiting additional information that a demonstrator or expert may have during training. It does so by developing a framework to imitate the “procedure’’ used by the expert to arrive at their decision of action, rather than imitating the expert action alone. This procedure is specified by a sequence of computations, that the framework estimates. They demonstrate the results of this framework on several different domains, and show it outperforms traditional behavior cloning based approaches to imitation learning.


**Questions:**

###  Primary Concerns:
1. Validity of access to expert chain of thought: The paper to a large extent is based on having few expert demonstrations, but with access to the entirety of the experts chain of thought. How reasonable is this assumption?

    a. When is expert chain of thought reasonable to assume access to? When is this infeasible?

    b. When the expert chain of thought is available readily (such as in the graph examples provided), isn’t it also feasible to generally set up a learner with an equivalent amount of knowledge at inference time (such as the ability to query a graph for traversability, expand a search, etc.)?

    c. When is it reasonable to assume expert chain of thought is easier to collect than simply additional demonstrations from the expert? In particular, if (state, action) pairs are easier to collect, would collecting more data that better spans the state-action space outperform PC?

2. Compounding errors of estimators: At inference time, the approach relies on predicted estimates of intermediate observations $x_l$, given only $s$, using a learnt estimator $p_{\theta}(x_l | x_{<l} ,s)$. This suggests that the approach suffers from the compounding error problem, where a small error in prediction of $x_0$ (more generally, $x_l$), would result in the input to the subsequent estimator being out of its training distribution. This in turn would lead to a (potentially larger) error in its prediction, eventually leading to potentially catastrophic error in prediction of final action $a$.

    a. To what extent does this problem occur in the proposed approach? What can be done to mitigate this?

    b. Would following a DAGGER like approach mitigate this?

3. Manually specified procedures: A key feature of the proposed approach is the procedure assumed to be used by the expert and the learner policy. This procedure is manually specified. This suggests the eventual imitation policy is a highly structured domain-specific policy, that is supervised at every intermediate level.  This marginally detracts from the applicability of the approach. That said, the introduction of such structure is acceptable, but requires discussion of the following points that is currently lacking -

    a. It is unclear how much effort is spent on specifying these procedures (particularly when it is unclear what the true chain of thought of the expert is), and how much PC benefits from this compared to the unstructured BC / AugBC approaches. For example, the BC / AugBC approaches are performing a much more difficult inference problem, estimating $p(a | s)$, whereas the individual inference problems that PC needs to perform ($p(a | x_L)$, $p(x_l | x_{l-1})$, $p(x_0 | s)$) are each much easier. How much does PC benefit from this? This is only briefly discussed in the limitations section.

    b.  When the true chain of thought is unavailable, or not easily approximated, what happens? This seems analogous to the problem of estimating the structure of a graphical model vs. inference on a given graphical model - perhaps the authors could provide directions of automatic inference of this chain of thought.

    c. If a significant effort is spent in specifying this procedure, could other baseline approaches benefit from equivalent domain-specific engineering work? For example, could a leaner agent on the graph traversal problem be provided with a graph structure, and the ability to query the graph for traversability, etc.? Would PC still outperform such a classical search?

4. Soundness of evaluation: It is unclear how fair some of the evaluations are. For example:

    a. Are the model capacities comparable across the PC, BC, AuxBC, AugBC approaches? Or does the PC approach have larger model capacity due to estimating the additional $p_{\theta}(x_l)$ estimators?

    b. It is also unclear if the parameterizations of PC vs BC / AuxBC approaches are entirely fair. For example, the BC and AuxBC models are parameterized with MLPs / CNNs in the maze traversal domain. In this case, the PC approach has inherent access to sequential / history information over time via the visited array, but AuxBC and BC approaches do not, since they are single-timestep models. What if the parameterizations were recurrent across all approaches, thus providing all approaches with temporal information? Does this help the AuxBC / BC approaches?

    c. It is unclear if the AuxBC approach is provided with sufficient or comparable information to the PC approach. Comparing AuxBC and PC aims to determine how beneficial procedural cloning as a central objective is compared to an auxiliary objective to behavior cloning. If the AuxBC approach does not have sufficient input information (such as search states, or object states in manipulation, that PC is provided in the form of supervision of the intermediate observations), it is not entirely fair to compare it against PC.

    d. It would also be interesting to see how much BC / AugBC benefit from having their intermediate layers supervised by an expert (say, with supervision in the form of feature matching losses on each layer of their architecture), and how PC compares to this.


###  Other concerns:
It is unclear when the $p_{\theta}$ estimators can be made auto-regressive vs. conditionally independent. Can this be made arbitrarily?
Line 206: It is unclear why the number of demonstrations dictates the conditional dependence of factorization.
The paper considers the cross-entropy objective; it is unclear if this is the best objective here. For example, one could also consider the joint distribution $p(s,a)$, and maximize the likelihood of the observed states and actions, with an according factorization.

###  Typos / minor edits:
Line 115: “many” fewer → far fewer
Line 120-121: Could be more specific about the “procedures” that directly influence actions through “computations”. Providing examples of these procedures and computations, their form, etc., would be useful.



**Limitations:**

### Limitations
The authors do discuss limitations of their paper briefly. However, the paper could benefit from a more thorough discussion of the limitations and their implications. For example -
1. How limiting is the manual specification of procedures? What problems would open up should this be resolved?
2. How limiting is the requirement of additional information from the expert? When is such information hard to access?
3. When is the cost of getting such additional information go beyond the cost of collecting additional demonstrations, thus making the problem setting irrelevant?

Another concern I have with the framing / claims of the paper is the language of learning the thought process / chain of thought of the agent used is very general, and allows for misattribution of the proposed frameworks capabilities. More precise language here could prevent adjacent communities from misunderstanding / overestimating the frameworks' capability.

###  Conclusion:
While there are several concerns voiced, I believe this paper introduces an interesting way of thinking about imitation learning problems where access to expert thought processes is reasonable. I believe this is a valuable contribution to the imitation learning community.




**Strengths And Weaknesses:**

### Pros:
1. Well motivated: The paper is well motivated. It discusses how traditional imitation learning approaches work, the drawbacks with such approaches, and how exploiting additional information that demonstrators / experts often have access to could improve imitation learning. It thus sets up the motivation for chain of thought imitation well. The paper, in general, is also well written and easy to follow, with minor exceptions (see below).
2. Sound and elegant problem set up: The paper sets up the problem well. The framing of learning the sequence of computations as a factored joint distribution is sound, and is an elegant (if simplistic) way of systematically representing these computations. Consequently, the presented chain of thought imitation is a fairly broadly applicable framework, and unifies similar ideas in various domains well in the imitation learning setting.
3. Performance on variety of tasks: The paper also does a good job evaluating on varied domains, and explaining the instantiations of its procedural imitation in each of these domains. The proposed method also outperforms classical imitation learning approaches on these domains, albeit with some concerns (see below).

For concerns / weaknesses and questions, please see below.

---

> ### Author Response · Authors · 2022-08-01
> **Response to Reviewer 4JKy (1/2)**
>
> Thank you for the insightful questions and comments!
>
> > 1 a) When is expert chain of thought reasonable to assume access to? When is this infeasible?
>
> PC is feasible when the expert is a planning, search, or multi-step algorithm that reveals additional information beyond the expert action. PC is infeasible when the expert is a black box that provides no additional information other than outputting an action when given a state. PC does not require the entirety of the expert’s chain of thought; e.g., access to some partial information about the MCTS procedure is enough for significant performance improvement, as shown in our MinAtar experiments.
>
> > 1 b)  Isn’t it also feasible to generally set up a learner with an equivalent amount of knowledge at inference time (such as the ability to query a graph for traversability, expand a search, etc.)
>
> For the BFS example, expert procedure information is only available on the set of training mazes, as running BFS requires annotations of wall locations which is not available on inference-time (test) mazes. We want navigation agents trained on a small set of annotated mazes to be able to generalize to new maps without annotations. Similarly, MCTS requires access to the simulator, which has privileged information useful for learning but will not be available when a policy is deployed in the real world.
>
> > 1 c) When is it reasonable to assume expert chain of thought is easier to collect than simply additional demonstrations from the expert? Would collecting more data that better spans the state-action space outperform PC?
>
> We want to point out that PC is not running additional programs to collect procedure data; the expert runs their procedure no matter what to compute the optimal action. PC simply records the procedure information that is often discarded under the current imitation learning paradigm. Collecting more state-actions pairs will also collect more procedure data.
>
> > 2 a) To what extent does compounding errors of estimators occur?
>
> The compounding error does exist, but since the multi-step model also receives more supervised signal to learn the intermediate steps from the additional procedure data, in practice the compounding error quickly becomes negligible as the amount of training data increases. We report the single-step and compounding accuracy during evaluation in [Figure 13](https://i.ibb.co/7vPyJDC/compounding-error.jpg) of the updated appendix.
>
> > 2 b) Would following a DAGGER like approach mitigate this?
>
> Yes - we believe following DAGGER would mitigate compounding errors exactly by the same mechanism as in traditional behavioral cloning.
>
>
> > 3 a) Efforts spent on specifying procedures and how much PC benefits from this compared to the unstructured BC / AugBC approaches
>
> PC operates under the situations where expert procedures are already (partially) given (e.g., a scripting policy, a known algorithm such as BFS or MCTS), so little effort is needed to manually specify the procedures. In other words, the expert itself breaks down $p(a|s)$ into $(p(a|x_L),... p(x_L|x_{L−1}), p(x0|s))$ during its computation, and PC imitates this decomposed computation step by step, as opposed to specifying the decomposition.
>
>
> > 3 b) When the true chain of thought is unavailable or not easily approximated, what happens?
>
> While being able to infer procedural information in hindsight when it is not available is out of scope of this work, we believe it may be possible to use latent variable models in such settings, where the latent state / actions can be are inferred without additional training data.
>
> > 3 c) Could other baseline approaches benefit from equivalent domain-specific engineering work?
>
> Instead of manually engineering domain specifications, PC utilizes domain-specific information when it is available, or learns to predict procedure observations autoregressively when domain specifications are not available (to mimick the procedure best to its ability given the available information). We can view PC as providing a way to incorporate domain-specific information in domain-agnostic imitation learning. Other ways of incorporating domain-specific information such as through graph structures as proposed by the reviewer can also be feasible, but in our experience do not lead to results as good as PC; see Figure 9, where we evaluate Value Iteration Networks (VIN), which leverage domain-specific engineering in gridworld.

---

> > ### Author Response · Authors · 2022-08-01
> > **Response to Reviewer 4JKy (2/2)**
> >
> > > 4 a) Model capacities across the PC, BC, AuxBC, AugBC.
> >
> > We ensured that the model capacities of all methods are comparable on the navigation and Bimanual sweep tasks. For MinAtar, we increased the CNN size of the baselines to account for the larger transformer parametrization of PC, which did not lead to improved baseline performance. In general, we did not find increasing the baselines’ model size helpful, as much of the baselines issue lies in overfitting / failure to generalize, as opposed to underfitting / insufficient model capacity.
> >
> > > 4 b) It is also unclear if the parameterizations of PC vs BC / AuxBC approaches are entirely fair… The PC approach has inherent access to sequential / history information over time via the visited array, but AuxBC and BC approaches do not, since they are single-timestep models.
> >
> > All methods are given the same number of $(s_t, a_t)$ pairs over time. While PC makes use of intermediate computations x_t, it does not have access to additional environment history (e.g., s_{t-1}) compared to BC / AuxBC; i.e., PC is not recurrent over time but rather recurrent over intermediate computation steps.
> >
> > > 4 c) It is unclear if the AuxBC approach is provided with sufficient or comparable information to the PC approach.
> >
> > AuxBC receives the same information as PC when computing the auxiliary loss, i.e., the visitation maps in maze navigation and the object states in manipulation. We will update the text to clarify this.
> >
> > > 4 d) How much BC / AugBC benefit from having their intermediate layers supervised by an expert.
> >
> > Thanks for the suggestion! We think extending PC to neural-network experts is an exciting direction for future work.
> >
> > > When the $p_\theta$ estimators can be made auto-regressive vs. conditionally independent.
> >
> > The best parametrization of $p_\theta$ depends on what procedure information is available for training PC. For example, in the BFS experiment where the visitation map that fully captures the search state is made available, conditional independence factorization would suffice, whereas in the MinAtar experiment where only parts of the MCTS procedure is available, autoregressive factorization is more desirable.
> >
> > > Line 206: It is unclear why the number of demonstrations dictates the conditional dependence of factorization.
> >
> > Autoregressive models need more data to train, so the conditional independence factorization is preferable if the procedure information available fully captures the computation state. We will update the text to clarify this.
> >
> > > Other alternative objective / factorizations.
> >
> > We are not modeling the joint distribution of $p(s, a)$ as the current state is given to a policy during evaluation. Other objectives / factorizations of PC (e.g., $p(s’, x_T, a, x_\pi, | s)$ where $x_T$ is some procedure data of the environment and $x_\pi$ is the procedure data of the expert) can be made possible.
> >
> > > Limitations.
> >
> > Thanks for these suggestions! We will include these limitations in our future manuscript. We want to additionally clarify that PC does not incur additional computations to collect procedure data, as the expert has to undergo the procedure to compute optimal actions anyways. PC simply records the procedure information that is otherwise discarded. The chain-of-thought rhetoric is our attempt at anthropomorphizing an autonomous agent, whereas procedure cloning denotes the precise method.

---

> > > ### Author Response · Authors · 2022-08-05
> > > **Follow-up to Reviewer 4JKy**
> > >
> > > Dear Reviewer, we wonder if we have answered all the questions you have about our work, or if there were any other questions that we can address. Thank you.

---

> > > ### Comment · Reviewer_4JKy · 2022-08-07
> > > **Response to Authors' Rebuttal**
> > >
> > > Thank you for the detailed response to my concerns. I think for the most part many of the clarifications I wanted have been provided. Points such as the applicability of the problem setting considered, and the details of what information is provided to various approaches, is useful clarification. I request the authors to provide this information in their text wherever possible. I believe the paper would benefit from having these points stated in the text (such as details about recurrence over steps rather than time, etc.).
> > >
> > > I would like to maintain my original rating of the paper.

---

### Official Review · Reviewer_mA33 · 2022-07-10

**Rating:** 4
**Confidence:** 3
**Soundness:** 3 good
**Presentation:** 3 good
**Contribution:** 2 fair

**Summary:**

Paper introduces a new paradigm for imitation learning -- procedure cloning. In addition to expert demonstrations, agents will also be provided with privileged information on how exactly the experts draw these plans, ex. the intermediate states of some search/planning algorithms. Then the learned policy will have to predict these states in a row and finally the action. The authors acknowledge some similar endeavors that happened before, which also leverage the extra information in RL and IL but in auxiliary tasks fashion. Experiments in various domains including path finding, robotic motion planning, and video game playing demonstrate the effectiveness of their proposed method, especially on zero-shot generalization.

**Questions:**

See [Weaknesses]

**Limitations:**

The authors mentioned the limitation on computation cost and the requirement of domain knowledge in applying their approach. However, I also find it fails to properly measure the extra overhead brought to experts as more information is yet to be required. It might not be a serious issue for some established algorithms, but as I mentioned above, we must consider what will happen when it is learning from humans; otherwise, the merit of this method could be on thin ice.

**Strengths And Weaknesses:**

[Strenghs]

+The paper is overall clear and well-written. The presentation is exceptionally illustrative and easy to follow. I like how figure 2 showcases their main contribution to prior arts in such a simple but efficient fashion.

+The idea is intuitive but reasonable. Although it is straightforward to me that more privileged information could lead to better performances, the authors did a good job comparing Aux BC and therefore justified the value of their proposed "procedure cloning".

+I like how they perform PC on atari with MCTS as an expert. Very well executed and the backward chaining style planning with GPT is impressive.

[Weaknesses]

Having said these above, I do have some concerns regarding the method itself and empirical results. I hope the author could clarify them in a rebuttal.

-The approach relies on the intermediate state of some other established algorithms to serve as the source of "thoughts". However, I'm wondering how can it learn from humans? The authors did mention this "where \pi* is a human demonstrator, we can ask the human to explain the thought process that led to their decision." but provide no results. All the experiments are about learning from some established algorithms, which I presume with perfect I.I.D. and even O.O.D. generalization on the selected domains. I do think the value of this method could be questionable without the proclaimed human case.

-Zero-shot generalization should be a central topic in the experiment part as claimed by the authors. But in general, it is not very well conducted and the results don't seem to be comprehensive, some of them are even less convincing. As far as I can tell, most of these zero-shot settings are simply created by using different random seeds, ex. new maze structures, novel initial states, etc, and there is no generalization test for the robotics planning domain. These generalization tests are still largely I.I.D. and comparably simple, considering the amount of extra information the proposed agents received. I will suggest re-designing the testing envs by following the protocol that aims at testing algorithmic generalization, ex. testing regimes proposed by [1, 2], where training and testing problems are distinctive on structures but still share the same primitives. In the pathfinding domain, it could be larger mazes during testing; For video games then it could be more opponents than training tasks. The results could be much more convincing if these challenging tests are included.

-Since the action is finally generated after predicting a few steps of "thoughts", it will be pivotal to also examine whether these "thoughts" are reasonable, or even interpretable, as they're trained to be so. However, I didn't find any examples in the main paper. Can the authors provide some of them in the rebuttal and the next revision of this manuscript? It will be better to also compare them with the prediction made by Aux BC.

[1] Alchemy: A benchmark and analysis toolkit for meta-reinforcement learning agents

[2] HALMA: Humanlike Abstraction Learning Meets Affordance in Rapid Problem Solving

---

> ### Author Response · Authors · 2022-08-01
> **Response to Reviewer mA33**
>
> Thank you for your helpful comments and questions. We hope our responses below address your concerns. Let us know if you have any further questions.
>
> > How can PC learn from humans?
>
> Scaling PC to incorporate human procedural information is an exciting next step for our work. We believe most of our experiments can easily adapt to human analogues. For example, rather than using a scripted policy in the robotic manipulation example, we can instead collect expert demos from human tele-operation and ask the humans to annotate their demonstrations with the trajectory – in the image – they were trying to follow. In other cases, natural language can serve as procedural information for non-NLP tasks (see Ahn 2022 and Huang 2022 for promising results in similar settings). Please let us know if you have suggestions for existing datasets with human-annotated procedure information, as we are interested in exploring this direction.
>
> > Zero-shot generalization evaluation.
>
> We respectfully disagree with the reviewer that our generalization evaluation is not comprehensive. For maze navigation, maze structures (i.e., dynamics) and initial positions are the most important components of an MDP over which generalization is often desirable (Mnih 2015, Nair 2015, Osband 2019, Zhang 2018). Using procedurally generated training and testing environments for evaluating generalization in RL is also a common practice. For instance, VIN (Tamar 2016) and Minigrid (Chevalier-Boisvert 2018) specifically test generalization across maze layouts.  For simulated robotics manipulation, the initial position of the particles are partitioned so that test-time configurations are never seen in training. This is the standard train/test split for this task and we chose to follow this protocol in order to be able to align and compare our results to existing work on this task. For MinAtar evaluation, enemies are spawned and move at a faster speed when we ramp up the game difficulty during evaluation, resulting in more enemies / harder games as suggested by the reviewer. The deterministic to stochastic environment generalization we evaluate on MinAtar is also common and realistic (Machado 2018). To further reassure the reviewer that PC generalizes well in the face of drastic distribution shift, we additionally provide results on maze navigation where the agent is trained on mazes with tunnel-shaped inner walls and evaluated in a zero-shot manner on mazes without inner walls or mazes with block-shaped inner walls. New results are presented in [Figure 15](https://i.ibb.co/NV7PjwK/empty-maze.jpg) of the updated appendix. PC exhibits much better generalization when the test mazes look drastically different from the training mazes.
>
> > Examine whether these "thoughts" are reasonable, or even interpretable.
>
> Thanks for the suggestion. [Figure 14](https://i.ibb.co/YQmfxx8/chains.jpg) in the updated appendix shows the visualization of PC’s learned BFS procedure.
>
> > Extra overhead brought to experts as more information is yet to be required
>
> We assume that the expert already undergoes the underlying procedures to compute the optimal action, and PC simply records the procedure information that is often discarded under the current imitation learning paradigm. We will include discussions on limitations where this assumption does not hold, which will indeed incur additional overhead as the review pointed out.
>
> References
> - InnerDialogue, language as additional supervision. Ongoing work. Suggestions? Human-based procedure datasets?
> - Huang, Wenlong, et al. "Inner Monologue: Embodied Reasoning through Planning with Language Models." arXiv preprint arXiv:2207.05608 (2022).
> - Ahn, Michael, et al. "Do as i can, not as i say: Grounding language in robotic affordances." arXiv preprint arXiv:2204.01691 (2022).
> - Mnih, Volodymyr, et al. "Human-level control through deep reinforcement learning." nature 518.7540 (2015).
> - Nair, Arun, et al. "Massively parallel methods for deep reinforcement learning." arXiv preprint arXiv:1507.04296 (2015).
> - Osband, Ian, et al. "Behaviour suite for reinforcement learning." arXiv preprint arXiv:1908.03568 (2019).
> - Zhang, Chiyuan, et al. "A study on overfitting in deep reinforcement learning." arXiv preprint arXiv:1804.06893 (2018).
> - Tamar, Aviv, et al. "Value iteration networks." Advances in neural information processing systems 29 (2016).
> - Chevalier-Boisvert, Maxime, Lucas Willems, and Suman Pal. "Minimalistic gridworld environment for openai gym." (2018).
> - Cobbe, Karl, et al. "Quantifying generalization in reinforcement learning." International Conference on Machine Learning. PMLR, 2019.
> - Cobbe, Karl, et al. "Leveraging procedural generation to benchmark reinforcement learning." International conference on machine learning. PMLR, 2020.
> - Machado, Marlos C., et al. "Revisiting the arcade learning environment: Evaluation protocols and open problems for general agents." Journal of Artificial Intelligence Research 61 (2018): 523-562.

---

> > ### Author Response · Authors · 2022-08-05
> > **Follow-up to Reviewer mA33**
> >
> > Dear Reviewer, we would like to ask if your concerns have been addressed by our additional generalization and visualization results, or if there were any other issues that would prevent you from increasing your score. Please let us know, and thank you for your time.

---

> > > ### Comment · Reviewer_mA33 · 2022-08-09
> > > **Thanks for the responses**
> > >
> > > I would like to thank the authors for their timely responses. My concerns about learning from humans and generalization tests are basically addressed. However, I do think the authors should elaborate more on the visualization part. In the revised version, only PC's learned BFS procedure is provided but as I suggested, methods that could possibly generate the same structure, ex. Aux BC, should be evaluated as well. More tasks are still yet to be covered. I hope these results could be available in their final version. Therefore I have to stick on with my original rating.

---

> > > > ### Author Response · Authors · 2022-08-09
> > > > **Follow-up to Reviewer mA33**
> > > >
> > > > Thank you for the follow-up. We will be sure to include a visualization of Aux BC's predictions in the final paper. We also believe that our evaluation across navigation, manipulation, and game play are comprehensive to demonstrate PC's advantage.

---

### Official Review · Reviewer_CmHn · 2022-07-11

**Rating:** 4
**Confidence:** 4
**Soundness:** 3 good
**Presentation:** 4 excellent
**Contribution:** 2 fair

**Summary:**

The paper attempts to introduce a new formulation to imitation learning. The key insight is that in order to achieve strong generalization, the imitation learner needs to model not only the action at a given state, but the computational procedure that determines the action. The authors name such learning problem procedure cloning (PC). The paper cites a typical 2D navigation problem as an example where the learner can benefit from PC: the procedure policy would effectively clone a BFS search procedure, including the cell expansion and backtracking steps. The paper proposes to model the PC problem using a transformer-like structure. During inference time, the procedures are predicted auto-regressively, similar to a seq-to-seq generation problem. The formulation is validated on four different types of domains ranging from 2D navigation to robot manipulation. The main findings are that PC is particularly helpful in low-data regime and out-of-distribution generalization thanks to its strong inductive biases.


**Questions:**

My main question is regarding the technical contribution of this paper given that there already exists a large body of works that has proposed various form of procedural cloning. See the full context in the main comment above.

**Limitations:**

I believe the paper has adequately addressed the limitation of the PC method.

**Strengths And Weaknesses:**

Overall I enjoyed reading the manuscript – the writing is superb. And I echo strongly with the message that imitation learning needs to go beyond modeling actions at a given state. I also appraise that the paper uses various types of domains to validate the main claim that modeling procedure is more advantageous than merely cloning actions.

However, as much as I like the key message of the paper, I do have strong doubts about the technical contributions of this paper, especially when there already exists a large body of work in imitation learning that has proposed various forms of procedural cloning.

The first thing that came to mind is the seminal work Neural Programmer-Interpreter (NPI) by Reed et al (2016). The paper proposes to learn the intermediate computations of any procedure that can be represented by a program-like structure. Similar to PC, the intermediate computation does not need to take any specific form – it can be a sorting algorithm or a policy that rotates a 3D model to its canonical view, hence the learning algorithm is domain-agnostic. And the model itself is also effectively a seq-to-seq architecture (LSTM as opposed to Transformer). A follow-up work of NPI (Xu et al., 2018) also shows that NPI can be used to perform long-horizon manipulation tasks where hierarchical imitation learning can be cast as a program induction problem.

The second type of algorithms are goal-conditional imitation learning (Mandlekar et al., 2020), or a more generalized form of Decision Transformers (DT). The PC formulation in Section 6.2 is effectively goal-conditional IL where the goal is expressed in goal points. There are many existing works that show that hierarchical or goal-conditional imitation learning methods show better sample efficiency and / or stronger generalization (Lynch et al., 2020) due to its inductive biases.

The third type of algorithms are learning-to-plan methods that clone the planning procedure itself. For example, Universal Planning Networks (Aravind et al., 2018) attempts to recover the planning procedure through a bi-level gradient-based optimization procedure. Similarly, various works have attempted to make planning faster by learning the stopping condition or plan feasibility.

One may argue that PC formulation may encompass all the works above. However, this argument runs the risk of overgeneralization, as any type of fully-supervised algorithms for decision making that exploits inductive biases can be considered as procedural cloning. And applying Transformer-like architectures to all the problems above mostly demonstrates the versatility of Transformers. Hence my main question for the author is: what’s the main contribution of this paper on top of existing works that leverage domain-specific supervisions to improve policy learning?

References:
- Reed, Scott, and Nando De Freitas. "Neural programmer-interpreters." arXiv preprint arXiv:1511.06279 (2015).
- Xu, Danfei, et al. "Neural task programming: Learning to generalize across hierarchical tasks." 2018 IEEE International Conference on Robotics and Automation (ICRA). IEEE, 2018.
- Chen, Lili, et al. "Decision transformer: Reinforcement learning via sequence modeling." Advances in neural information processing systems 34 (2021): 15084-15097.
- Mandlekar, Ajay, et al. "Iris: Implicit reinforcement without interaction at scale for learning control from offline robot manipulation data." 2020 IEEE International Conference on Robotics and Automation (ICRA). IEEE, 2020.
- Lynch, Corey, et al. "Learning latent plans from play." Conference on robot learning. PMLR, 2020.
- Srinivas, Aravind, et al. "Universal planning networks: Learning generalizable representations for visuomotor control." International Conference on Machine Learning. PMLR, 2018.

---

> ### Author Response · Authors · 2022-08-01
> **Response to Reviewer CmHn**
>
> We thank the reviewer for helpfully pointing out a large body of related work. We provide our response regarding each referenced work below, these important discussions are added to the Appendix D of the paper. Please let us know if you have any remaining concerns or questions!
> > Neural Programmer-Interpreter (NPI) by Reed et al (2016) and follow-up work of NPI (Xu et al., 2018)
>
> While NPI may be interpreted to apply to any “program” similar to PC, its demonstrated applications (both in Reed et al and Xu et al) are based on problems that exhibit hierarchical and modular solutions with shared and repeated subprograms (e.g., a “pick” primitive within a block-stacking task). Thus, NPI not only advocates to imitate the full program but also parameterize the agent in a modular way – like a program with function calls and return values – to take advantage of this modular and repeated structure. The main argument in these existing papers hinges on the modular decomposition of both task and agent, which allows for more efficient data sharing, especially in multiple-task settings. In contrast, our PC evaluations are on environments with much less modular structures, and we avoid using specialized agent parameterizations in favor of more generic architectures (e.g., transformers). Thus, we believe our work is complementary to NPI, showing that the paradigm of using procedural information – while first proposed by the NPI work to some extent – applies to much more general settings than initially suggested by NPI.
>
> > Goal-conditional imitation learning.
>
> While some of our experiments may be interpreted as utilizing “goals” as intermediate computation, there is a key difference from goal-conditioned imitation learning. Namely, goal-conditioned imitation learning advocates for using portions of the observation (or learned functions of the observation) as goals, whereas PC uses information beyond what is available in the observation (e.g., coordinate positions of objects). Thus, the common argument in goal-conditioned imitation centers around getting more learning signal (goal-reaching) from the same data (s,a,s’ tuples). In contrast, PC shows that having richer data in the demonstrations (procedure information) is useful, *even if that data is not available to the agent during inference*. We also note that another common argument made in goal-conditioned imitation learning is attributing their sample efficiency benefits to reduced temporal frequency induced by the hierarchical design, whereas PC shows benefits without any change in temporal frequency. Thus, we believe that PC presents a novel approach.
>
> > Learning-to-plan methods.
>
> Learning-to-plan methods (e.g., Universal Planning Networks, Value Iteration Networks) are distinct from PC in that they train policies end-to-end using state-action tuples. No additional supervision of the intermediate computations of the expert is used; rather, these methods effectively propose a different policy parameterization leveraging inductive biases (i.e., with an embedded end-to-end differentiable planner). As shown in our evaluations of Implicit BC (Figure 5) and Value Iteration Networks (Appendix B.1), flexible policy parametrizations and architectures with desired inductive bias still fail to generalize without proper integration of procedure information during training, showing that PC provides benefits orthogonal to these existing works.
>
> These important points the reviewer has raised are very helpful in situating PC in relation to prior work, which we include in Appendix D of the updated paper. We note that PC is not “merely applying Transformers to imitation learning”, as PC focuses on the setting different from traditional imitation learning, and Transformer is only one modeling choice of PC when procedure information is partially available.

---

> > ### Author Response · Authors · 2022-08-05
> > **Follow-up to Reviewer CmHn**
> >
> > Dear Reviewer, we would like to ask if your concerns around the related work and novelty of our method have been addressed, or if there were any other issues that would prevent you from increasing your score. Please let us know, and thank you for your time.

---

### Official Review · Reviewer_13Cs · 2022-07-26

**Rating:** 7
**Confidence:** 3
**Soundness:** 3 good
**Presentation:** 2 fair
**Contribution:** 3 good

**Summary:**

This paper proposes to go beyond the conventional observations->actions setup of imitation learning, and instead learn a more meaningful mapping of observations - > procedure (intermediate states/obs) -> actions. The hypothesis is that this harder prediction problem results in agents learning a more understanding of the task at hand, rather than action level mimicking. The authors draw parallels to chain of thought prompting in NLP literature, and perform a series of experiments in navigation, manipulation, and discrete control tasks to demonstrate the efficacy of procedure cloning.

**Questions:**

* Can you explain the key idea of procedure cloning could be explained in a simpler manner (see above).
* This is not a requirement, but I would encourage the authors to try their method of harder experiments which are currently not feasible to solve with Behaviour Cloning alone. Improved metrics on simple tasks is fine, but what new capabilities does procedure cloning give us? Some ideas are trying on MineRL or StarcraftUnplugged.

**Strengths And Weaknesses:**

Strengths:
* A very elegant and intuitive idea that should be generally applicable to any RL domain, provided procedure demonstrations are available.
* The experiments cover a wide range of domains, and are not limited to particular setting. There seems to be clear benefit of using Procedure Cloning over standard Behaviour Cloning.

Weaknesses:
* Section 4 needs a lot of re-writing to explain the key idea (which is pretty simple but has been very tersely explained). In particular, I think having a Python like pseudocode would make it very easy for readers to quickly grasp the idea and re-implement. The section relies on un-necessary notation and graphical models when simply {inputs, outputs, loss function, architecture} would have worked.

---

> ### Author Response · Authors · 2022-08-01
> **Response to Reviewer 13Cs**
>
> > The key idea of procedure cloning could be explained in a simpler manner.
>
> Thanks for the suggestions! We will include a pseudocode (copied below) in Section 4 to improve readability.
>
> **Inputs** Datasets $D_\Pi= \{(s^{(i)},\textbf{x}^{(i)}, a^{(i)})\}_{i=1}^{n}$, parametric policy $p\_{\theta,\phi,\psi}(a, {\bf x} | s)$, learning rate $\eta$
>
> **for** $t = 1, ..., T$ **do**
>
> &nbsp; Sample batch $\{(s^{(i)},\textbf{x}^{(i)}, a^{(i)})\}_{i=1}^{B}$ from $D_\Pi$
>
> &nbsp; Compute loss $J_\mathrm{PC}(\phi,\theta,\psi) = \hat E_{\{(s^{(i)},\textbf{x}^{(i)}, a^{(i)})\}_{i=1}^{B}}[-\log p\_{\theta,\phi,\psi}(a, \textbf{x}|s)]$
>
> &nbsp; Update $\theta\leftarrow \theta + \eta\nabla_{\theta}J_\mathrm{PC}$, $\phi\leftarrow \phi + \eta\nabla_{\phi}J_\mathrm{PC}$, and $\psi\leftarrow \psi + \eta\nabla_{\psi}J_\mathrm{PC}$
>
> **end for**; **return** $p\_{\theta,\phi,\psi}$
>
> > Harder experiments which are currently not feasible to solve with Behaviour Cloning alone
>
> We emphasize that BC already struggles to solve the existing set of environments. For example it achieves 0% accuracy in maze navigation (orange line in Figure 4) and 0 reward in the harder MinAtar games such as Seaquest and SpaceInvaders (orange line in Figure 7). We are continuing to scale up PC to even larger-scale games and environments as part of ongoing work, and MineRL and StarcraftUnplugged are great test beds.

---

> > ### Author Response · Authors · 2022-08-05
> > **Follow-up to Reviewer 13Cs**
> >
> > Dear Reviewer, we wonder if your concerns around the simplicity of our presentation has been addressed in the author response. Thank you.

---

### Meta-Review · Area_Chair_mFgY · 2022-08-27

**Recommendation:** Accept
**Confidence:** Less certain

**Metareview:**

All reviewers acknowledged the rebuttal or replied to the author responses.
The paper proposes to clone procedures in imitation learning rather than blindly mimicking actions. One reviewer is still not convinced that the experiments are sufficient, while I believe they are solid enough for a conference paper. The relation to other methods that take a similar conceptual approach on a high-level is now clear, but should be incorporated in the main paper.

**Award:**

No

---

### Decision · Program_Chairs · 2022-09-14

Accept